# The Expression Pattern of Bcl-2 and Bax in the Tumor and Stromal Cells in Colorectal Carcinoma

**DOI:** 10.3390/medicina58081135

**Published:** 2022-08-21

**Authors:** Nenad Kunac, Natalija Filipović, Sandra Kostić, Katarina Vukojević

**Affiliations:** 1Department of Pathology, Forensic Medicine and Cytology, University Hospital Centre Split, 21000 Split, Croatia; 2Department of Anatomy, Histology and Embryology, School of Medicine, University of Split, 21000 Split, Croatia

**Keywords:** colorectal carcinoma, bcl-2, bax, stromal cells

## Abstract

*B**ackground and objectives:* The epithelial and stromal tissues both play a role in the progression of colorectal cancer (CRC). The aim of this study was to assess the expression of anti-apoptotic Bcl-2 and pro-apoptotic Bax in the epithelium as well as the lamina propria of normal colonic controls, low-grade tumor samples and high-grade tumor samples. *Materials and Methods:* A total of 60 samples consisting of both normal colonic and carcinoma samples was collected from the Department of Pathology, Cytology and Forensic Medicine, University Hospital Center, Split from January 2020 to December 2021. The expression of Bcl-2 and Bax markers was semi-quantitatively and quantitatively evaluated by recording immunofluorescence stain intensity and by counting stained cells in the lamina propria and epithelium. Analysis of positive cells was performed using the Mann–Whitney test. *Results:* In all samples, Bcl-2 was significantly more expressed in the lamina propria when compared with the epithelium. Bax was significantly more expressed in the epithelium of normal and low-grade cancer samples when compared with their respective laminae propriae. The percentage of Bcl-2-positive cells in lamina propria is about two times lower in high-grade CRC and about three times lower in low-grade CRC in comparison with healthy controls. Contrary to this, the percentage of Bax-positive cells was greater in the epithelium of low-grade CRC in comparison with healthy control and high-grade CRC. *Conclusions:* Our study provides a new insight into Bcl-2 and Bax expression pattern in CRC. Evaluation of Bcl-2 expression in the lamina propria and Bax expression in the epithelium could provide important information for colorectal cancer prognosis as well as potential treatment strategies.

## 1. Introduction

Colorectal cancer (CRC) is the third most common malignancy and the second leading cause of cancer-related mortality in the world, with an estimated number of 1.8 million new cases and about 881,000 deaths worldwide in 2018 [1,2]. Apoptosis was the first characterized phenomenon of programmed cell death. When adverse events occur within the cell, Bax promotes apoptosis by undergoing a conformational change, membrane insertion and oligomerization, thereby providing a channel for cytochrome C to exit outside the mitochondria. This in turn activates caspase-mediated cell death [3].

The balance of pro- and anti-apoptotic proteins may determine cell fate. However, the discovery of more proteins interacting with Bcl-2, the identification of non-apoptotic functions of certain members of the Bcl-2 family, and the discovery of other non-apoptotic functions of Bcl-2 have challenged this simplistic model [4].

Apoptosis is crucial for the well-being of cells and plays a role in development of the organism, homeostasis of tissues, metabolism, immunity, and eradication of unnecessary cells from the body. The extrinsic pathway (also known as the death receptor pathway) involves death receptors bound to the cell surface along with their respective ligands. The intrinsic or mitochondrial pathway is exclusively governed by proteins of the Bcl-2 family. Therefore, the Bcl-2 family proteins are attractive candidates for anti-cancer therapeutic strategies [5].

The levels of Bcl-2 expression vary across cell types and tumor cell lines. Additionally, each member of the Bcl-2 family is allocated a special task in apoptosis and their function could also be determined by the cell type [6]. Apoptosis inhibition is the critical mechanism by which cancer persists. Cells with abnormal genetic components bypass apoptosis and become immortal tumors. Bcl-2 drives apoptosis by suppressing Bax function. It has been documented that Bax inhibition can resist apoptosis and respond to different chemotherapeutic drugs [6]. The employment of Bax as a marker for colorectal cancer prognosis or prediction has been met with uncertainty. This is due to conflicting reports from different studies [7,8,9] showing that both an increase and a decrease in Bax expression are associated with tumor survival. This poses a challenge when considering chemotherapeutic strategies that target Bax.

In colorectal cancer, the intestinal epithelium’s homeostasis is interrupted due to disharmony in mitotic and apoptotic processes. Bcl-2 is said to reduce the cell’s metabolic activity and thereby makes it resistant to apoptosis [10].

The current standard prognosis for colorectal cancer is the estimation of extent of tumor, which is assessed by the depth of the tumor invasion, and the involvement of the lymph nodes. However, the current prognostic methods have been met with discrepancy [11]. Therefore, identification of biological markers of collateral cancer is the aim of alternative prognostic methods.

Communication between cells and their immediate environment is an important part of maintaining both normal tissues as well as promoting tumor growth. Tumor cells and their accompanying stroma communicate and may cause disease and metastasis. This phenomenon is therefore important in the prognosis of cancer [12].

In this study, we aim to assess the expression of anti-apoptotic Bcl-2 and pro-apoptotic Bax, to elucidate a new prognostic method, taking into account epithelial as well as lamina propria tissue apoptosis.

## 2. Experimental Procedures

### 2.1. Normal and Carcinogenic Adult Human Tissues

All tissues used in this study were processed after obtaining permission from the Ethical and Drug Committee of the University Hospital in Split. Both normal and colonic tissues were classified as postmortem material according to the guidelines provided by the Declaration of Helsinki. Evaluation of the tumor samples was conducted by macroscopic quantification. A total of 60 samples consisting of both normal colonic and tumor samples was collected from the Department of Pathology, Cytology and Forensic Medicine, University Hospital Center Split from January 2020 to December 2021 (see the Appendix A). Within these samples, twenty were low-grade colonic cancers and twenty were high-grade colonic cancers that were grade classified according to WHO (World Health Organization) guidelines (Figure 1). For the control group, colonic mucosa from the autopsy samples of twenty deceased adult patients who did not have any gastrointestinal lesions were used. All tissue samples were fixed using 4% paraformaldehyde in phosphate buffer and dehydrated in a series of graded ethanols. Samples were then embedded in paraffin, serially sectioned at 5 μm thickness, and mounted on glass slides. 

### 2.2. Immunofluorescence Staining

Normal colonic, low-grade tumor and high-grade tumor samples were selected. The epithelium and lamina propria were stained for both anti-apoptotic Bcl-2 and pro-apoptotic Bax, and the staining intensity was analyzed. Positively stained cells were then quantified in the two regions of interest the epithelial tissue and the lamina propria (Table 1).

After deparaffinization of the samples in xylene and rehydration in ethanol and water, sections were treated in a microwave oven at 95 °C for 12 min in sodium citrate buffer (pH 6.0). Briefly, sample sections were incubated overnight with a combination of Bcl-2 antibody (1:50; M0887, DAKO, Glostrup, Germany) and Bax antibody (1:500; Chemicon International, Inc., Temecula, CA, USA). Slides were then washed with PBS and incubated for one hour with an appropriate combination of secondary antibodies Alexa Fluor 488 Donkey Anti-Rabbit (1:500; A-11058 Invitrogen Molecular Probes Inc., Eugene, OR, USA) and TRITC Goat Anti-Mouse IgG (1:500; ab6786, Abcam, Cambridge, UK). Nuclei were then counterstained with DAPI, and sections were placed under a coverslip (Immuno-Mount, ThermoShandon, Pittsburgh, PA, USA). Microscopic analysis of the sections was then performed using an Olympus BX61 (Tokyo, Japan) microscope equipped with a DP71 camera (SPOT Insight, Diagnostic Instruments, USA) using the Olympus Cell A software and assembled using Adobe Photoshop (Adobe Systems, San Jose, CA, USA). 

### 2.3. Semi-Quantification

The staining intensity of the control colonic mucosa and low- and high-grade CRC tissue samples were semi-quantitatively measured and allocated to four groups, according to the intensity of the stain: no intensity = −; mild intensity = +; moderate intensity = ++; and strong intensity = +++ (Table 2).

### 2.4. Quantification of Bcl-2- and Bax-Positive Cells

The cell counting was performed in the two regions of interest (lamina propria and epithelium). The stained cells were counted as positive, and the unstained cells were counted as negative. Images were captured using a digital camera (DP71 Olympus, Tokyo, Japan) mounted on an Olympus BX51 microscope. Image J software (National Institutes of Health, Bethesda, MD, USA) was used to divide each area of the images into squares of 20 × 20 µm at a magnification of 100×. Positive and negative cells were counted as described previously. The percentage of positive cells in both the lamina propria and epithelium was calculated and expressed as mean ± SD. Significance of the data was analyzed using the Mann–Whitney test. *p* values < 0.05 were considered statistically significant.

### 2.5. RNA Isolation and qRT-PCR

The Sigma-Aldrich protocol for DNA isolation by affinity chromatography was used with the GenElute™ FFPE RNA Purification Kit. The formalin-fixed paraffin-embedded (FFPE) samples were deparaffinized through a series of xylene and ethanol washes. Digestion process was performed with Proteinase K and Digestion Buffer A. Buffer RL and ethanol were added to the lysate and applied onto a spin-column. After the RNA isolation, the High-Capacity cDNA Reverse Transcription Kit by Applied Biosystems was used for reverse transcription. A master mix with cDNA and selected forward and reverse primers (Table 3), SYBR green and nuclease free water was mixed and added to a 96-well plate. All samples were analyzed in duplicate. RPS9 was used as a housekeeping gene. The samples were analyzed by Applied Biosystems™ 7500 Real-Time PCR Systems.

## 3. Results

### 3.1. Semi-Quantification of Bcl-2 and Bax Staining in the Lamina Propria and Epithelial Tissues

In the lamina propria of the control and low-grade samples, a strong expression of Bcl-2 was identified. In the lamina propria of the high-grade samples, moderate expression of Bcl-2 was observed (Table 1).

In the epithelium of all samples, Bcl-2 was mildly expressed. With respect to Bax expression, the epithelial tissue of low-grade samples displayed a strong staining intensity, whereas the control and high-grade samples displayed mild expression (Table 2).

In the lamina propria, all samples displayed mild expression (Figure 2).

### 3.2. The Lamina Propria Displays a Significant Expression of Bcl-2 in Comparison with Epithelial Tissues

Figure 3a,d shows that there was approximately 70% of BCL-2-positive cells in the lamina propria of control samples compared with approximately 5% positive cells in the epithelium (Mann–Whitney, *p* < 0.0001). There were approximately 20% Bcl-2-positive cells in the lamina propria of low-grade samples compared with approximately 2% expression in the epithelium (Mann–Whitney, *p* < 0.001). Bcl-2 expression in the lamina propria of high-grade samples was approximately 40% of total cells in comparison with approximately 2% expression in epithelial tissues (Mann–Whitney, *p* < 0.001).

### 3.3. The Epithelium Displays a Significant Expression of Bax in the Low-Grade Samples in Comparison with the Lamina Propria

The epithelium of control samples displayed approximately 8% Bax-positive cells compared with approximately 2% of positive cells in the lamina propria (Mann–Whitney, *p* < 0.01) (Figure 2a,c). The epithelium of low-grade samples displayed approximately 45% Bax-positive cells compared with approximately 2% Bax-positive cells in the lamina propria (Mann–Whitney, *p* < 0.0001). The epithelium of high-grade samples displayed about 5% Bax-positive cells compared with approximately 2% Bax-positive cells in the lamina propria, with no significant differences between epithelial and lamina propria tissues.

Of note, there was no co-expression of Bcl-2 and Bax in the same tissues (Figure 3c,d).

The percentage of Bcl-2-positive cells in lamina propria is about two times lower in high-grade CRC and about three times lower in low-grade CRC in comparison with healthy controls. Contrary to this, the percentage of Bax-positive cells was striking in the epithelium of low-grade CRC in comparison with healthy controls and high-grade CRC.

The qPCR analysis confirmed the above-mentioned results of Bcl-2 and Bax quantification (Figure 4). 

## 4. Discussion

Cancers generate heterogeneous, complex environments that contain multiple cells that proliferate, including both cancer cells and stromal cells along with the extracellular matrix (ECM1); this either directly or indirectly contributes to a tumor cell’s maintenance [13]. The stroma of normal tissues plays the part of an organ support structure, which is vital for sustaining the tumor microenvironment within cancer, since it provides both support and nutrients to these cells [14]. This is because these cells need not only nutrients, but also metabolite withdrawal and gas exchange for them to grow, which is partly provided by blood vessel circulation; therefore, there is an elevated angiogenesis rate in tumors, as this enables the cells to continually proliferate [13].

Apoptosis refers to a form of programmed cell death that is necessary for tissue homeostasis. Within tumor initiation, a contributing factor is the de-regulation in the balance between apoptosis and proliferation, which is seen more often in the colon, as apoptosis is a vital process in intestinal turnover; in fact, when apoptosis is inhibited, it subsequently leads to tumor progression and transformation [15]. Key regulators of apoptosis are the BCL-2 family of proteins, as they have been implicated in not only colorectal cancer initiation, but also its progression and resistance to therapy [15]. 

The tumor microenvironment (TME) plays a crucial role in tumor progression, therapeutic response and patient outcomes [16]. TME has a dynamic composition, including various cell types, such as cancer-associated fibroblasts (CAFs), tumor-associated macrophages (TAMs), regulatory T cells (Tregs) and myeloid-derived suppressor cells (MDSCs), as well as extracellular factors that surround cancer cells [17]. The tumor microenvironment provides inappropriate signals that lead to the maintenance of tumorigenesis, tumor progression and cancer therapy resistance [18]. Furthermore, by evaluating preferential patterns of metastatic dissemination to certain organs, it can be observed that the TME provides support for tumor occurrence and progression. In fact, a combination of the colonizing cancer cells’ adaptability and a favorable environment is essential [19].

Previous studies have indicated that evaluating ratios of Bax and Bcl-2 may provide more useful insights on the prognosis of colorectal cancer, the patient’s lifespan and response to therapeutic drugs [8,9]. The current study aimed at finding a stand-alone method of evaluating colorectal cancer prognosis.

Our study on the comparison of Bcl-2 and Bax expression in epithelial and lamina propria tissues in normal, low-grade and high-grade carcinoma showed a significant number of positive cells expressing the anti-apoptotic marker Bcl-2 in the lamina propria of all samples. The strong expression of Bcl-2 in the lamina propria was also additionally semi-quantitatively confirmed by immunofluorescence staining data, which detected a strong Bcl-2-positive cell population in all samples, with normal and low-grade cancer tissues displaying strong intensity. Additionally, the pro-apoptotic marker Bax was significantly expressed in the epithelial tissues of control and low-grade samples. Semi-quantitative immunofluorescence staining also concurs with these data, wherein a strong BAX expression was observed in the epithelial tissues of low-grade cancer samples. Interestingly, there was no significant co-expression of both the pro- and anti-apoptotic markers in the same tissue.

It is a well-known fact that the normal epithelial tissue becomes a pre-malignant entity known as an adenoma that finally turns into a malignant structure known as carcinoma [20]. This carcinoma can then metastasize or spread to cells and tissues in its vicinity and may ultimately spread to other organs in the system. However, only about 5% of adenomas ever turn malignant. Our data display an approximately 20% higher anti-apoptotic Bcl-2-positive cell population in the lamina propria of high-grade samples compared with the lamina propria of low-grade samples, accounting for the differences in the pre-malignant and malignant cancer states. Interestingly, Bcl-2 was only mildly expressed in the epithelium of all samples. Therefore, a reduction in the number of Bcl-2-positive cells in the epithelium could imply cancer invasion from the epithelium to the lamina propria. Additionally, checking for significant increases in the number of Bax-positive cells in the epithelium in comparison with the lamina propria, as displayed in the low-grade samples in our study, would provide an important prognostic for evaluating cancer.

The lamina propria plays an essential role in epithelial transformation. It has been shown that growth index increases in the lamina propria of precancerous tissues and eventually increases further in the lamina propria of colorectal cancers [21]. These findings could explain significant expression of anti-apoptotic Bcl-2 in the lamina propria of low-grade samples and subsequent significant expression in high-grade samples when compared with epithelial tissues in our study. Additionally, these findings can also account for the significant reduction in pro-apoptotic Bax in the lamina propria of low-grade samples. 

As Nocquet et al. [10] explains apoptosis’ intrinsic pathway is dependent upon MOMP, as it leads to caspase activation and the resultant cell integrity loss. For cancer cell survival, it is imperative that there be mitochondrial apoptosis resistance that takes place either up or downstream of MOMP. These cancer cells interact with cancer-associated fibroblasts (CAFs), which play a part in both tumor progression and growth, not to mention the tumor’s survival and invasion [22]. CAFs also act as protectors for cancer cells, keeping them from being damaged by chemotherapies or pro-apoptotic drugs, as they modulate the mitochondrial apoptosis-related proteins. When there is the oxidative stress of DNA damage, BAX and/or BAK oligomerization can be initiated at the mitochondrial membrane, leading to MOMP and subsequently, cytochrome c’s cytosolic release; this stimulates caspase activation and apoptosis [10].

In other words, stress stimuli change the equilibrium between pro- and anti-apoptotic BCL-2 family proteins that play a part in regulating BAX and BAK oligomerization. The anti-apoptotic proteins BCL-2, BCL-xL and MCL-1 prevent MOMP through directly interacting with the pro-apoptotic proteins, protecting cancer cells from stress stimuli caused by chemotherapies [23]. Furthermore, CAFs prefer chemoresistance through regulating some BCL-2 family anti-apoptotic protein levels in malignant cells [23]. CAFs contribute to colorectal cancer development, with many originating from local pericryptal fibroblasts [24]. When there is a balance between apoptosis and proliferation, there is homeostasis in the colon. 

This reduction in pro-apoptosis Bax is also in agreement with the classical apoptosis-related model of the colorectal carcinoma sequence consisting of normal mucosa to adenoma to carcinoma.

The tumor–stroma ratio has been shown to be a critical prognostic for colorectal cancer. Therefore, epithelial stromal interactions can be investigated as candidates for anti-cancer therapies. Targeting Bcl-2 specifically in the lamina propria of the cancer tissue could prove to be a potential therapeutic option.

Furthermore, detecting Bcl-2 specifically in the lamina propria of samples may provide a useful prognosis, as a form of evaluation of stromal-related apoptosis.

The surprising result in this study was that lower anti-apoptotic Bcl-2 was found in the cancerous tissues compared with the normal controls. This was unexpected because a defining feature of cancerous cells is their resistance to undergo apoptosis. However, the sensitivity of cancerous cells to apoptotic signals may not be related to their level of apoptosis. Additionally, it has been shown that apoptosis levels decrease with an increase in the TNM stage of the cancer, implying a decrease in the cells’ response to apoptotic signals [24].

Deregulated apoptosis is observed in most tumor types and is considered one of the hallmarks of cancer. Evasion of apoptosis allows tumor cells to bypass oncogene-induced cell death and can also promote sustained tumor growth, survival during metastatic spread and therapy resistance. De-regulation of the BCL-2 family not only occurs during tumorigenesis and outgrowth but is also observed as part of the tumor evolution that takes place in response to therapy [23].

Not surprisingly, an increased expression of pro-survival BCL-2 proteins is found in several cancer types. CRC develops in a stepwise manner with sequential accumulation of specific genetic mutations that dictate the progression from adenoma-to-carcinoma stages [25]. This progression is accompanied by several changes in the apoptotic threshold of the cancer cells with an overall inhibition of apoptosis [26]. Most members of the BCL-2 family show altered expression patterns in CRC tumors, which plays a role in cancer progression and therapy resistance.

The correlation between levels of Bcl-2 expression and clinical outcome is inconsistent across several studies [27,28,29,30]. However, these studies may vary according to number of patients, patient age, stage of the disease, etc. 

As with most studies, the design of the current study is subject to limitations. In our research, we did not use additional methods to differentiate tumor epithelial cells and stromal cells, but we determined the location of the immunofluorescent signal using standard histopathological morphology. Hints for epithelial differentiation include epithelioid cells (round to oval cells) with nesting arrangement and/or glandular and cribriform formations surrounded with desmoplastic stroma with feeding vessels separating tumor cells. 

Additional differentiation with the help of diagnostic methods such as immunohistochemistry or immunofluorescence is of course possible, but the poorly differentiated and high-grade malignant phenotype can vary, and positivity can be generally weak and focal. In addition, the intra-tumor heterogeneity can make interpretation more difficult [31].

Our study considered epithelial transformation and the interaction of the epithelium with the stromal tissues indicating decreased Bax expression in primary tissues compared with metastatic tissues. The crucial role of Bax/Bcl-2 ratio has been enumerated in a few studies. It has been suggested that low levels of Bax/Bcl-2 can lead to increased resistance to apoptosis. Further, low Bax/Bcl-2 ratios may result in poor prognosis and an increase in cancer invasion. Interestingly, our study displayed a decreased Bax/Bcl ratio in the low-grade epithelial tissues, which could imply that the cancer may be spreading to the lamina propria.

Several members of the BCL-2 family protein are deregulated upon transformation, with colon adenomas having greater levels of anti-apoptotic BCL proteins [32]. There is a possibility that different chemotherapeutic agents may affect the expression level and clinical roles of molecules that are involved in apoptosis, especially Bax and Bcl-2 [33]. There are also numerous changes that occur in the cancer cell’s apoptotic threshold that, overall, work to prevent apoptosis; additionally, most members of the BCL-2 family exhibit changed expression patterns in colorectal carcinoma tumors, having a significant impact on both cancer progression and therapy resistance [15]. However, future studies confirming this potential are needed.

In conclusion, the present study provides a novel insight into the role of Bax/Bcl-2 ratios in epithelial–stromal tissues that might have implications in colorectal carcinoma prognosis. 

## Figures and Tables

**Figure 1 medicina-58-01135-f001:**
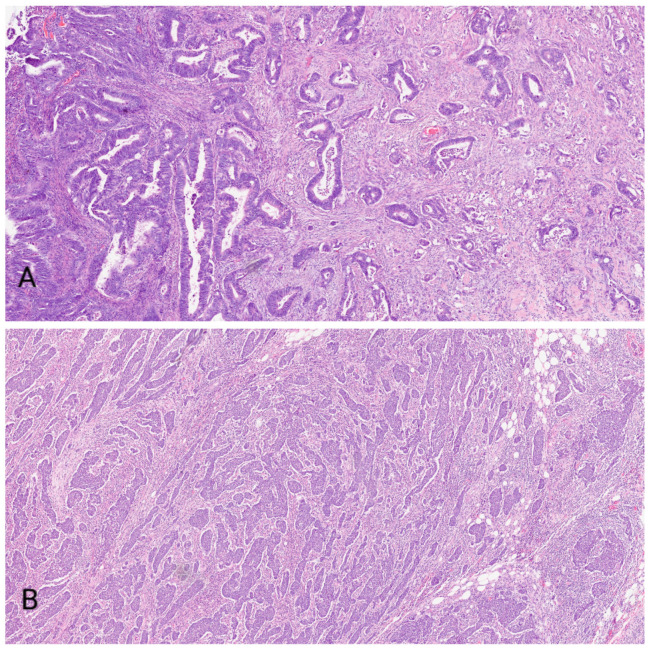
Figure shows low-grade (**A**) and high-grade (**B**) colorectal cancer.

**Figure 2 medicina-58-01135-f002:**
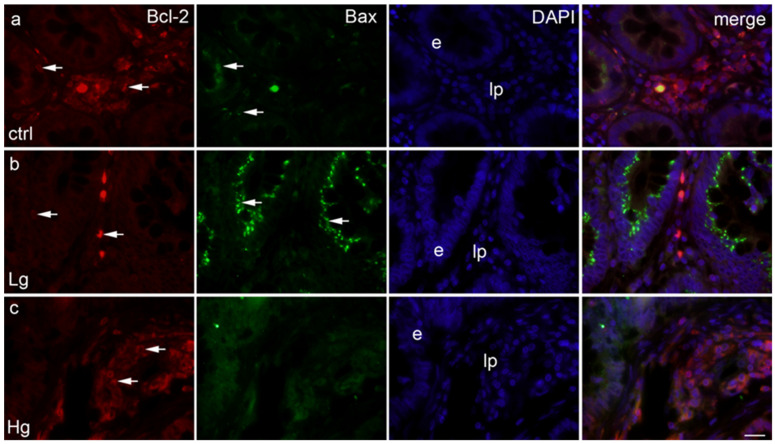
Section through the healthy human colon tissue (ctrl) (**a**), section through the human low-grade colon cancer (Lg) (**b**), section through the human high-grade colon cancer (Hg) (**c**); positive cells (arrows), lp–lamina propria, e-epithelium. Triple immunofluorescence staining for Bcl-2 (red), Bax (green) and DAPI (blue), scale bar 10 μm.

**Figure 3 medicina-58-01135-f003:**
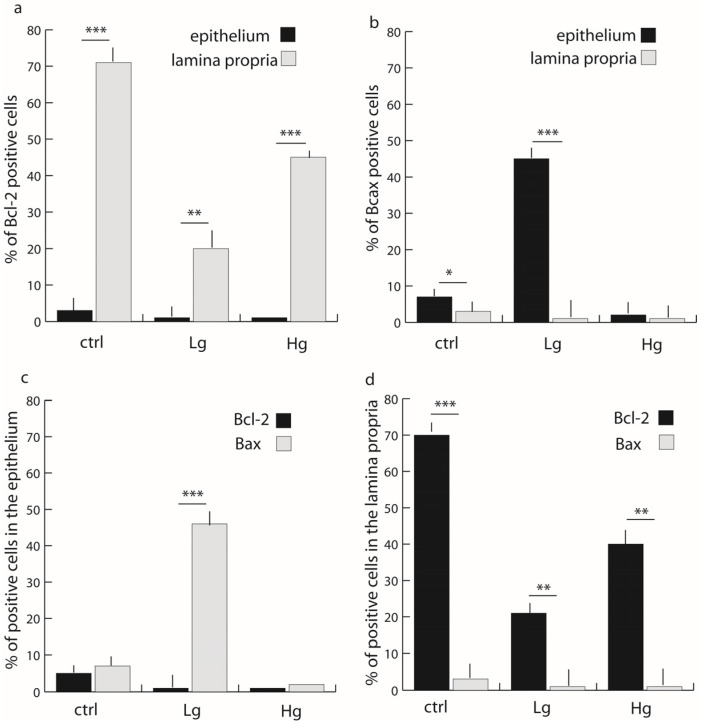
Distribution of Bcl-2- (**a**), and Bax- (**b**) positive cells in the lamina propria (lp) and epithelium (e), and distribution of Bcl-2- and Bax-positive cells in the epithelium (**c**) and lamina propria (**d**) in control colon tissue (ctrl), low-grade colon cancer (Lg), and high-grade colon cancer (Hg). Data are presented as mean ± SD. Significant differences (Mann–Whitney test) are indicated by * *p* < 0.01, ** *p* < 0.001, *** *p* < 0.0001.

**Figure 4 medicina-58-01135-f004:**
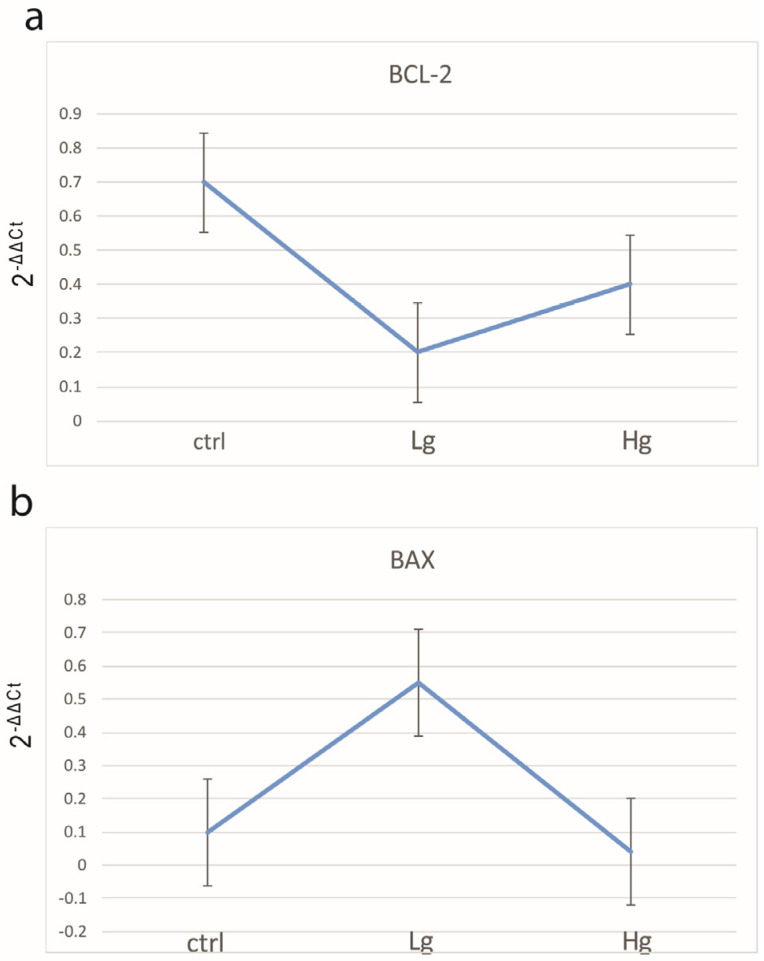
The qRT-PCR analysis of colorectal samples and controls using primers for BAX and BCL2 and their calculated 2^−^^ΔΔ^^Ct^. Number of samples analyzed were 5 samples per group. Control colon tissue (ctrl), low-grade colon cancer (Lg), and high-grade colon cancer (Hg).

**Table 1 medicina-58-01135-t001:** Quantification of the positively stained cells in the two regions of interest in the epithelial tissue and the lamina propria of the samples and control group.

**Bcl-2**	**ctrl**	**Lg**	**Hg**
e	3	1	1
lp	71	20	45
**Bax**	**ctrl**	**Lg**	**Hg**
e	7	45	2
lp	3	1	1

**Table 2 medicina-58-01135-t002:** Staining intensity of Bcl-2 and Bax in the control human colon and CRC (low-grade and high-grade).

Antibody	Bcl-2			Bax		
Structure/Group	ctrl	Lg	Hg	ctrl	Lg	Hg
e	+	+	+	+	+++	+
lp	+++	+++	++	+	+	+

**Table 3 medicina-58-01135-t003:** Primers used in qRT-PCR.

Gene	Forward Primer	Reverse Primer
*BAX*	5′-TCA GGA TGC GTC CAC CAA GAA G-3′	5′-TGT GTC CAC GGC GGC AAT CAT C-3′
*BCL2*	5′-GTG GAT GAC TGA GTA CCT GAA C-3′	5′-GCC AGG AGA AAT CAA ACA GAG G-3′
*RPS9*	5′-GGA TTT CTT AGA GAG ACG CCT G-3′	5′-GGA CAA TGA AGG ACG GGA TG-3′

## Data Availability

The authors have already attached Appendix A.

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
