# Peer review of "The Expression Pattern of Bcl-2 and Bax in the Tumor and Stromal Cells in Colorectal Carcinoma"

_medicina, 2022, doi:10.3390/medicina58081135_

Round 1

Reviewer 1 Report

Kunac et al looked at the expression of Bcl2 and BAX in normal colonic, low grade and high grade samples. They found differential expression pattern of these components using immunofluorescence and report it here. While the study is of interest, I have few major concerns.

1. The sample size of 5 samples per condition is not representative enough. The authors need to evaluate more samples.

2. Currently the quantitation depends on just one technique. Can the authors use qPCR from tissue sections to get an overall idea. I understand that the challenge might be to isolate parts of the section, but there are techniques out there.

3. Bcl2 being anti apoptotic, I would expect it to be high in high grade, as it might be needed to resist apoptosis, but it is interesting to see the opposite. Some explanation around why this is the case, would help strengthen the significance of the research.

Author Response

Response to reviewer #1 comments:

Kunac et al looked at the expression of Bcl2 and BAX in normal colonic, low grade and high grade samples. They found differential expression pattern of these components using immunofluorescence and report it here. While the study is of interest, I have few major concerns.

  1. The sample size of 5 samples per condition is not representative enough. The authors need to evaluate more samples.

Response: Thank you for your comment. We agree that the sample size is small. Therefore, we increased the numbers of samples in all groups to 20 samples per group. We added necessary changes into Material and Method sections and re-evaluate all the quantitative data and statistics.

  1. Currently the quantitation depends on just one technique. Can the authors use qPCR from tissue sections to get an overall idea. I understand that the challenge might be to isolate parts of the section, but there are techniques out there.

Response: Thank you for your comment. We agree with your comment and we have undertaken the additional analyses with qPCR, but we were able to perform analyses only on 5 samples in each group due to limited resources.

  1. Bcl2 being anti apoptotic, I would expect it to be high in high grade, as it might be needed to resist apoptosis, but it is interesting to see the opposite. Some explanation around why this is the case, would help strengthen the significance of the research.

Response: Thank you for your comment. In the "discussion" section, we have added the section below.

De-regulated apoptosis is observed in most tumor types and is considered one of the hallmarks of cancer. Evasion of apoptosis allows tumor cells to bypass oncogene-induced cell death and can also promote sustained tumor growth, survival during metastatic spread and therapy resistance. De-regulation of the BCL-2 family not only occurs during tumorigenesis and outgrowth but is also observed as part of the tumor evolution that takes place in response to therapy [Maji, S.; Panda, S.; Samal, S.K.; Shriwas, O.; Rath, R.; Pellecchia, M.; Emdad, L.; Das, S.K.; Fisher, P.B.; Dash, R. Bcl-2 Antiapoptotic Family Proteins and Chemoresistance in Cancer. Adv Cancer Res 2018;137:37-75.]

Not surprisingly, an increased expression of pro-survival BCL-2 proteins is found in several cancer types. CRC develops in a stepwise manner with sequential accumulation of specific genetic mutations that dictate the progression from adenoma-to-carcinoma stages [Fearon, E.R.; Vogelstein, B. A genetic model for colorectal tumorigenesis. Cell 1990 Jun 1;61(5):759-67.]. This progression is accompanied by several changes in the apoptotic threshold of the cancer cells with an overall inhibition of apoptosis [Bedi, A.; Pasricha, P.J.; Akhtar, A.J.; Barber, J.P.; Bedi, G.C.; Giardiello, F.M.; Zehnbauer, B.A.; Hamilton, S.R.; Jones, R.J. Inhibition of apoptosis during development of colorectal cancer. Cancer Res 1995 May 1;55(9):1811-6.]. Most members of the BCL-2 family show altered expression patterns in CRC tumors, which plays a role in cancer progression and therapy resistance.

Reviewer 2 Report

The topic of the manuscript is entitled ‘The expression pattern of Bcl-2 and Bax in the tumor and stromal cells in colorectal carcinoma’’. The authors aimed to evaluate the Bcl-2 & Bax expression in lamina propria and epithelium in CRC. This manuscript needs revisions.

My specific comments are appended as below:

1) In the introduction, Lane 30-35, the authors have given the 2008 prevalence of colorectal carcinoma. The authors should give the latest CRC prevalence. 

2) In Lane 64-66, the authors should give reference

3) The authors may need to give clinical characterization (tumor size, stage/tumor grade cutoff..etc) of the samples in the supplementary section.

4) In Figure 1, positive cells with – lamina propria, - epithelium identification is not clear, The Bcl-2 not only express lamina propria and it expresses other cell types (Ex: Enteric glial cells). The authors should do another Immunohistochemistry analysis for better identification of lamina propria and epithelium.

Author Response

Response to reviewer #2 comments:

The topic of the manuscript is entitled ‘The expression pattern of Bcl-2 and Bax in the tumor and stromal cells in colorectal carcinoma’’. The authors aimed to evaluate the Bcl-2 & Bax expression in lamina propria and epithelium in CRC. This manuscript needs revisions.

My specific comments are appended as below:

1) In the introduction, Lane 30-35, the authors have given the 2008 prevalence of colorectal carcinoma. The authors should give the latest CRC prevalence.

Response: Thank you for your comment. We added latest reference.

2) In Lane 64-66, the authors should give reference

Response: Thank you for your comment. We added the missing reference.

3) The authors may need to give clinical characterization (tumor size, stage/tumor grade cutoff..etc) of the samples in the supplementary section.

Response: Thank you for your comment. We provided the supplementary table with the clinical data.

4) In Figure 1, positive cells with – lamina propria, - epithelium identification is not clear, The Bcl-2 not only express lamina propria and it expresses other cell types (Ex: Enteric glial cells). The authors should do another Immunohistochemistry analysis for better identification of lamina propria and epithelium.

Response: Thank you for your comment. We honestly did not consider doing additional immunohistochemical analysis of stromal cells because the histological samples and immunofluorescence samples were reviewed by three histologists and two pathologists with many years of experience in the morphology of normal intestinal mucosa and colorectal cancer samples.

Round 2

Reviewer 1 Report

The authors have addressed my concerns and I am happy to recommend acceptance

Author Response

Response to reviewer #1 comments:

The authors have addressed my concerns and I am happy to recommend acceptance,

Response: Thank You for your useful comments. Spelling errors have been corrected.

Reviewer 2 Report

I thank the authors for providing the response. My major question on IHC still remains unanswered and it is not clear to me how authors differentiate between the Epithelial-stromal cells. This raises the question of the credibility of the work performed.

Author Response

Response to reviewer #2 comments:

I thank the authors for providing the response. My major question on IHC still remains unanswered and it is not clear to me how authors differentiate between the Epithelial-stromal cells. This raises the question of the credibility of the work performed.

Response: Thank you for your comment and we fully understand Your reservations. For this reason, we propose the following paragraph at the end of the discussion section.

As with the majority of studies, the design of the current study is subject to limitations. In our research, we did not use additional methods to differentiate tumor epithelial cells and stromal cells, but we determined the location of the immunofluorescent signal using standard histopathological morphology. Hints for epithelial differentiation include epithelioid cells (round to oval cells) with nesting arrangement and/or glandular and cribriform formations surrounded with desmoplastic stroma with feeding vessels separating tumor cells.

 Additional differentiation with the help of diagnostic methods such as immunohistochemistry or immunofluorescence is of course possible, but the poorly differentiated and high-grade malignant phenotype can vary, and positivity can be generally weak and focal. In addition, the intra-tumor heterogeneity can make interpretation more difficult [Selves J, Long-Mira E, Mathieu MC, Rochaix P, Ilié M. Immunohistochemistry for Diagnosis of Metastatic Carcinomas of Unknown Primary Site. Cancers (Basel) 2018 Apr 5;10(4):108.].

Also, in the case of our research quantitative image analysis should be used to generate high-content data through application to a technique known as multiplexing, which allows co-expression and co-localization analysis of multiple markers in situ with respect to the complex spatial context of tissue regions, including the stroma, tumor parenchyma, and invasive margin [Baxi V, Edwards R, Montalto M, Saha S. Digital pathology and artificial intelligence in translational medicine and clinical practice. Mod Pathol 2022 Jan;35(1):23-32.]. The current advances in digital pathology offer practical advantages, including enhanced accuracy and precision, the ability for digital images to be uploaded and reviewed by multiple pathologists, and the acquisition and processing of large and complex datasets which can be the subject of further research.

Round 3

Reviewer 2 Report

I appreciate the author's response. However, it still partially answers my concern. In case authors are not in a position to do an additional experiment, should clearly show the morphological picture with a brightfield/ HE image (I believe authors have retained the slides!). Additionally, the justification given in lines 342-349 does not matches my question. 

Author Response

Response to reviewer #2 comments:

I appreciate the author's response. However, it still partially answers my concern. In case authors are not in a position to do an additional experiment, should clearly show the morphological picture with a brightfield/ HE image (I believe authors have retained the slides!). Additionally, the justification given in lines 342-349 does not matches my question.

Response: Thank you for your comment and suggestion. The authors are not in a position to do an additional experiment (immunofluorescence labeling of the epithelial and stromal cells), so in the manuscript we attach an example of a histological image of high- and low-grade colorectal adenocarcinoma.

Regarding the text in lines 342-349, we only thought to give an overview of possible future research because we currently have a VENTANA DP 200 slide scanner in our department and we are in the process of acquiring algorithms that could read multiplex immunohistochemical and immunofluorescent staining. We fully agree that it does not answer your question and we will withdraw the text in the specified lines.
